# Lossy Micromaser Battery: Almost Pure States in the Jaynes–Cummings Regime

**DOI:** 10.3390/e25030430

**Published:** 2023-02-27

**Authors:** Vahid Shaghaghi, Varinder Singh, Matteo Carrega, Dario Rosa, Giuliano Benenti

**Affiliations:** 1Center for Nonlinear and Complex Systems, Dipartimento di Scienza e Alta Tecnologia, Università degli Studi dell’Insubria, via Valleggio 11, 22100 Como, Italy; 2Istituto Nazionale di Fisica Nucleare, Sezione di Milano, via Celoria 16, 20133 Milano, Italy; 3Center for Theoretical Physics of Complex Systems, Institute for Basic Science (IBS), Daejeon 34126, Republic of Korea; 4CNR-SPIN, via Dodecaneso 33, 16146 Genova, Italy; 5Basic Science Program, Korea University of Science and Technology (UST), Daejeon 34113, Republic of Korea; 6NEST, Istituto Nanoscienze-CNR, 56126 Pisa, Italy

**Keywords:** quantum energy storage, ergotropy, steady states, quantum thermodynamics

## Abstract

We consider a micromaser model of a quantum battery, where the battery is a single mode of the electromagnetic field in a cavity, charged via repeated interactions with a stream of qubits, all prepared in the same non-equilibrium state, either incoherent or coherent, with the matter–field interaction modeled by the Jaynes–Cummings model. We show that the coherent protocol is superior to the incoherent one, in that an effective pure steady state is achieved for generic values of the model parameters. Finally, we supplement the above collision model with cavity losses, described by a Lindblad master equation. We show that battery performances, in terms of stored energy, charging power, and steady-state purity, are slightly degraded up to moderated dissipation rate. Our results show that micromasers are robust and reliable quantum batteries, thus making them a promising model for experimental implementations.

## 1. Introduction

The description of the dynamics of open quantum systems via system–reservoir collision models [1,2,3,4,5,6,7,8,9,10,11,12,13,14,15,16] is rapidly spreading in a variety of research areas, see [17,18] for reviews. Notably, in the growing field of quantum thermodynamics [19,20,21,22,23], it has been applied to address fundamental problems such as the relaxation to equilibrium [3,4,5,6], the relationship between information and thermodynamics [9], the efficiency of thermodynamic heat engines [10,11,12], exposed to either thermal or non-equilibrium reservoirs [8,12,14], the relevance of non-Markovian effects [7,12], and strong system–bath coupling [10].

Recently, collision models have been used to describe the charging process of quantum batteries [13,14,15,16,24,25], that is, of quantum mechanical systems suitable to store energy in some excited states [15,16,26,27,28,29,30,31,32,33,34,35,36,37,38,39,40,41,42,43,44,45,46,47,48,49,50,51,52,53,54], to be later released on demand. The charging of the battery, modeled by a quantum harmonic oscillator or a large spin, via sequential interactions (collisions) with a stream of qubits, has exhibited enhanced performances with respect to classical counterpart, when the qubits are prepared in some coherent state, with respect to the classical, incoherent charging [15,16].

Moreover, it has been possible to show that by this charging protocol, and when the incoming qubits have a certain degree of coherence, the cavity reaches an effectively steady state which is both highly excited and essentially *pure*. Interestingly, these features extend beyond the weak coupling regime, and are present both in the strong and ultrastrong coupling regimes of field–matter interaction [25]. Achieving a pure state is appealing in terms of ergotropy [55,56,57], that is, of the amount of energy that can be extracted from a battery via unitary operations, and while for a mixed state part of the excitation energy (measured from the ground state energy of the quantum battery) cannot be used, for a pure state it is always possible to reach the ground state by a suitable unitary operation. Therefore, for a pure state, the ergotropy equals the mean excitation energy. Moreover, the fact that an effective steady state is achieved forbids the cavity to absorb an unbounded amount of energy, a possibility that would be dangerous, conceptually equivalent to the burning of an overcharged classical battery.

In this paper, we show first of all that the battery stability is obtained only in the case of coherent charging, while in the incoherent case slight deviations from the ideal, fine-tuned conditions, can effectively burn the battery. Moreover, the reported excellent properties, in the coherent case, of a micromaser as a quantum battery call for a deep analysis of its stability in the presence of decoherence mechanisms such as quantum noise [18,58,59,60,61]. Here, we consider the Jaynes–Cummings regime [62], where counter-rotating terms in the qubit-cavity collisions are neglected. Such approximation is naturally valid in the weak coupling regime, in which dissipative effects are small during the time of a collision and the qubit-cavity coupling strength *g* is much smaller than the cavity frequency ω. Moreover, it can be pushed to the strong or ultra-strong coupling regime, 0.1≲g/ω≲1, provided a simultaneous frequency modulation for both the qubit and the field is implemented [63]. We will model battery dissipation via a standard Lindblad master equation approach [64,65,66], with losses due to the finite cavity lifetime. While dissipation naturally avoids overcharging problems, forbidding unbounded energy absorption, its impact on the effectiveness of the charging process, and in particular on the steady-state purity, is a question to be carefully considered. In this paper, we show that the steady-state energy and purity are rather weakly affected up to $\gamma t_{r}=0.1$, with γ being the cavity decay rate and $t_{r}$ the time interval between two consecutive collisions. This stability analysis, and the consequent results, represent a crucial and necessary step in view of possible future experimental implementations of micromaser quantum batteries.

The paper is organized as follows. In Section 2, we describe the collision model here investigated. Incoherent and coherent charging protocols are discussed in Section 3 and Section 4, respectively. In Section 5, we consider the effects of dissipation, focusing on the coherent case, which is advantageous in the ideal, noiseless case. Finally, our conclusions are drawn in Section 6.

## 2. The Model

The quantum battery (QB) that we consider consists of a quantized electromagnetic field in a cavity, which for our purposes can be modeled as a quantum harmonic oscillator. The QB is initially prepared in the ground state, |0〉, of the harmonic oscillator. Energy is then accumulated by sending into the cavity a stream of qubits sequentially (or two state systems), interacting with the harmonic oscillator and transferring energy into it. The initial state of each qubit reads
(1)$$\begin{array}{cc} \rho_{q} & =q|g\rangle \langle g|+\left(1-q)\right|e\rangle \langle e|+c\sqrt{q(1-q)}\left(|e\rangle \langle g|+|g\rangle \langle e|\right)\;, \end{array}$$
where |*g*〉 is the ground state of the qubit while |*e*〉 represents the excited state. The parameters *q* and *c* control level populations and the degree of coherence, respectively. The interaction between a qubit and the cavity field is described by the Hamiltonian [62]
(2)$$\begin{array}{cc} \hat{H} & ={\hat{H}}_{0}+{\hat{H}}_{1}\;,\\ {\hat{H}}_{0} & \equiv \omega_{F}{\hat{a}}^{†}\hat{a}+\frac{1}{2}\omega_{q}{\hat{\sigma }}_{z}\;\\ {\hat{H}}_{1} & \equiv g\left(\hat{a}{\hat{\sigma }}_{+}+{\hat{a}}^{†}{\hat{\sigma }}_{-}+{\hat{a}}^{†}{\hat{\sigma }}_{+}+\hat{a}{\hat{\sigma }}_{-}\right)\;, \end{array}$$
where $\omega_{q}$ and $\omega_{F}$ are the frequencies of the qubit and the field, respectively; ${\hat{a}}^{†}$ and $\hat{a}$ are the creation/annihilation operators of the harmonic oscillator and ${\hat{\sigma }}_{z}$, ${\hat{\sigma }}_{+}$, and ${\hat{\sigma }}_{-}$ denote the number, creation, and annihilation operators for the qubit, respectively. Finally, *g* is the coupling constant for the interaction. We work in units such that ℏ=1.

To describe the dynamics, it is convenient to move to the interaction picture. By further assuming that the qubits and the field are in *resonance*, $\omega_{q}=\omega_{F}\equiv \omega$, the Hamiltonian simplifies to
(3)$${\hat{H}}_{I}=g\left(\hat{a}{\hat{\sigma }}_{+}+{\hat{a}}^{†}{\hat{\sigma }}_{-}+e^{i2\omega t}{\hat{a}}^{†}{\hat{\sigma }}_{+}+e^{-i2\omega t}\hat{a}{\hat{\sigma }}_{-}\right)\;.$$

The time evolution of the qubit–battery system is governed by the time-evolution operator
(4)$${\hat{U}}_{I}\left(\tau \right)\equiv T\exp\left(-i\int_{0}^{\tau }{\hat{H}}_{I}\left(t\right)dt\right)\;,$$
where T is the time-ordering operator, and τ denotes the interaction time between a single incoming qubit and the harmonic oscillator. The QB state after each collision can then be obtained by tracing out the qubit degrees of freedom.

As it is well-known, under the weak-coupling regime—$\frac{g}{\omega }\ll 1$—Equation (Equation 4) effectively reduces to the Jaynes–Cummings evolution operator,
(5)$${\hat{U}}_{I}=\exp\left(-i\theta \left(\hat{a}{\hat{\sigma }}_{+}+{\hat{a}}^{†}{\hat{\sigma }}_{-}\right)\right),$$
where we have introduced the parameter θ≡gτ.

In the context of quantum batteries [37,38,52], we are interested in fast charging and discharging, which requires going beyond the regime $\frac{g}{\omega }\ll 1$, while Equation (Equation 5) is sic et simpliciter no longer valid beyond this regime, we can, nevertheless, extend its validity into the strong coupling regime $0.1≲\frac{g}{\omega }≲1$. For that purpose, we take advantage of a simultaneous external frequency modulation of the qubit and field frequencies, as described in [63]. By modulating the frequencies we modify the Hamiltonian of the model to:(6)$$\begin{array}{cc} \; & \hat{H}\to {\hat{H}}^{\prime }={\hat{H}}_{0}^{\prime }+{\hat{H}}_{1}^{\prime }\;,\\ \; & {\hat{H}}_{0}^{\prime }\equiv \left(\omega_{F}+\eta \nu \cos\left(\nu t\right)\right){\hat{a}}^{†}\hat{a}+\frac{1}{2}\left(\omega_{q}+\eta \nu \cos\left(\nu t\right)\right){\hat{\sigma }}_{z}\;,\\ \; & {\hat{H}}_{1}^{\prime }\equiv {\hat{H}}_{1}\;, \end{array}$$
where η and ν are the modulation amplitude and frequency, respectively. By a frequency modulation, the time evolution operator becomes modified from Equation (Equation 4) to Equation (Equation 5), which is therefore also valid in the ultrastrong coupling regime.

In the Jaynes–Cummings framework, the evolution of the battery state, $\rho_{B}$, can be described by the following master equation relating the state of the battery after k+1 collisions to the state of the battery after *k* collisions:(7)$$\begin{array}{cc} \rho_{B}\left(k+1\right) & =\left(1-q\right){\hat{c}}_{N+1}\rho_{B}\left(k\right){\hat{c}}_{N+1}+q{\hat{s}}_{N+1}\hat{a}\rho_{B}\left(k\right){\hat{a}}^{†}{\hat{s}}_{N+1}+\left(1-q\right){\hat{a}}^{†}{\hat{s}}_{N+1}\rho_{B}\left(k\right){\hat{s}}_{N+1}\hat{a}+q{\hat{c}}_{N}\rho_{B}\left(k\right){\hat{c}}_{N}\\ \; & +ic\sqrt{q(1-q)}\left[{\hat{c}}_{N}\rho_{B}\left(k\right){\hat{s}}_{N+1}\hat{a}+{\hat{c}}_{N+1}\rho_{B}\left(k\right){\hat{a}}^{†}{\hat{s}}_{N+1}-{\hat{s}}_{N+1}\hat{a}\rho_{B}\left(k\right){\hat{c}}_{N+1}-{\hat{a}}^{†}{\hat{s}}_{N+1}\rho_{B}\left(k\right){\hat{a}}^{†}{\hat{c}}_{N}\right]\;, \end{array}$$
where the operators ${\hat{c}}_{N+1}$, ${\hat{c}}_{N}$, and ${\hat{s}}_{N+1}$ stand for $\cos\left(\theta \sqrt{\hat{N}+\hat{1}}\right)$, $\cos\left(\theta \sqrt{\hat{N}}\right)$, and $\frac{\sin\left(\theta \sqrt{\hat{N}+\hat{1}}\right)}{\sqrt{\hat{N}+\hat{1}}}$, respectively.

The incoherent protocol corresponds to c=0, where the qubits are in a fully mixed state and without any degree of coherence. The other limiting case, c=1 (The effect of a complex phase in the coherence parameter, i.e., c∈C, has been considered at length, both in the Jaynes–Cummings as well as in presence of counter-rotating terms, in [25]. For the Jaynes–Cummings dynamics, the presence of a phase factor is completely irrelevant and can be neglected), corresponds to the fully coherent protocol, where the battery is charged by a stream of qubits prepared in a pure state, with the superposition of the states |g〉 and |e〉.

## 3. Incoherent Charging Protocol

Let us first focus on the case in which the incoming qubit is given by an incoherent mixture, i.e., in which c=0. In this case, Equation (Equation 7) simplifies to
(8)$$\begin{array}{cc} \rho_{n}\left(k+1\right) & =|s_{n+1}{|}^{2}\left(q\rho_{n+1}\left(k\right)-\left(1-q\right)\rho_{n}\left(k\right)\right)+{\left|s_{n}\right|}^{2}\left(\left(1-q\right)\rho_{n-1}\left(k\right)-q\rho_{n}\left(k\right)\right)+\rho_{n}\left(k\right)\;, \end{array}$$
where $\rho_{n}\left(k\right)$ denotes the diagonal elements of the system density matrix, $\rho_{nn}\left(k\right)$, and $s_{n}\equiv \sin\left(\theta \sqrt{n}\right)$.

The main feature to observe in Equation (Equation 8) is that the dynamics turns out to be peculiar when the parameter θ takes appropriate (fine-tuned) values of the form $\theta =\frac{\pi }{\sqrt{m}}$, with *m* being a positive integer. To see this, let us consider first the case in which all the qubits are in the excited state, i.e., q=0, and study the equation defining the steady state, $\rho_{n}\left(k+1\right)=\rho_{n}\left(k\right)$ for each *n*. It reads
(9)$$|s_{n+1}{|}^{2}\rho_{n}\left(k\right)-{\left|s_{n}\right|}^{2}\rho_{n-1}\left(k\right)=0\;\;\forall n.$$

When θ is fine-tuned to satisfy $\theta =\pi /\sqrt{m}$, the steady state, $\rho_{n}^{s.s.}$, turns out simply to be
(10)$$\rho_{n}^{s.s.}=\delta_{n,m-1}\;,$$
which means that the steady state is a *pure state* and, more precisely, it is a number state. We will refer to this property as *trapping*. In such a situation, the amount of extractable work from the battery turns out to be deterministic and maximal, since the ergotropy coincides with the total energy for pure states [55]. The possibility of building number states out of a micromaser has already been discussed long ago in the literature [67]. Here, we use such an opportunity to build an incoherent quantum battery having the wanted properties of reliable and maximal work extraction.

To the purpose of building an effective device out of this property, one needs to be sure that such steady state can be reached in a finite (ideally small) number of collisions (thus increasing the charging power). Moreover, it is crucial that such a trapping property is robust, meaning that it is not lost for deviations of the parameter θ from the fine-tuned value $\theta =\pi /\sqrt{m}$, and for q≠0.

We start by considering the fine-tuned case at q≠0: It can be shown that the steady-state equations are solved as follows:(11)$$\begin{array}{cc} \; & \rho_{n}^{s.s.}=r^{n}\rho_{0}^{s.s.}\;,\;\;\forall n<m\;,\\ \; & \rho_{n}^{s.s.}=0\;,\;\forall n\geq m\;,\\ \; & \rho_{0}^{s.s.}=\frac{r-1}{r^{m+1}-1}\;, \end{array}$$
where we defined $r\equiv \frac{1-q}{q}$. From Equation (Equation 11) we see that the trapping property is essentially preserved for q≠0 and θ fine-tuned, since the energy of the steady state remains well-concentrated around the wanted value, $E_{m-1}$, obtained for q=0. Given the steady state Equation (Equation 11), we can compute the purity, P, of such state, defined as $P\equiv \mathrm{Tr}\left(\rho^{2}\right)=\sum_{n}\rho_{n}^{2}$, where the last equality uses the fact that in the incoherent case ρ is diagonal. We obtain
(12)$$P^{s.s.}=\frac{r-1}{r+1}\frac{{\left(r^{2}\right)}^{m+1}-1}{{\left(r^{m+1}-1\right)}^{2}}\;,$$
which, for sufficiently large *m*, can be approximated by $P^{s.s.}\approx 1-2q$. Hence, we conclude that the purity property, enjoyed by the steady-state for q=0, is not stable and it is lost for q≠0.

On the other hand, we see that the trapping property itself is not stable for deviations of θ from its fine-tuned value. In such a case, the steady-state equations do not admit any trapping solution and the battery levels are populated without any upper bound. In Figure 1, we show the energy stored in the battery, as a function of the number of collisions for both fine-tuned values of the θ and *q* parameter and for small deviations around their fine-tuned values. We find, in agreement with the previous analytical discussion, that the battery, when θ and *q* are fine-tuned, reaches the steady state |*m*〉. On the other hand, we also observe that the system is not robust to small perturbations of θ. In such cases, a temporary, *metastable*, state is reached at first, for longer times as θ approaches a fine-tuned value, followed by a further evolution which continues indefinitely. We believe that such deviations of θ from its fine-tuned values introduce an effective random noise to the steady-state fine-tuned populations, determined via Equation (Equation 9), which become relevant at time $t^{☆}\propto 1/{\left(\Delta \theta \right)}^{2}$ (here, Δθ denotes the difference between the actual value of θ and its closest fine-tuned value), an estimate consistent with the inset of Figure 1. In addition, for fine-tuned values of θ but q≠0, the trapping properties are robust and the energy of the steady state is close to its fine-tuned counterpart. This latter property emphasizes that the two parameters, *q* and θ, controls two completely different aspects of the charging performance of the micromaser: *q* is responsible for the purity of the steady state, while θ is what controls the very existence of a steady state.

Since fine-tuned values of the θ parameter lead to stable energies, we have studied in these cases the purity of the battery state ρ(k), as a function of the number of collisions. The results are reported in Figure 2. As expected, in the case of fine-tuned θ values, the final state has purity 1 when q=0, but such a property is lost for q≠0. As we will see in the next section, this drop in purity by reducing the degree of population inversion is completely avoided when the incoming qubits have a certain amount of purity, i.e., when c≠0. This property represents one of the main motivations to consider the coherent charging protocol as advantageous.

In summary, we have shown that in the case of incoherent charging, the fine-tuned steady states |*m*〉 are fragile against small perturbations of both the θ and *q* parameters.

## 4. Coherent Charging Protocol

For completeness and ease of comparison with the incoherent case, let us now consider the case in which the qubits are coherent, i.e., c≠0 in Equation (Equation 7), which was the subject of Ref. [25]. We will focus on the case c=1, which means that the incoming qubits are in a superposition state: (13)$$\begin{array}{c} \rho_{q}=|\psi \rangle \langle \psi |\;,\;|\psi \rangle \equiv \sqrt{q}|g\rangle +\sqrt{1-q}|e\rangle \;. \end{array}$$

In this case, it has already been discussed in the literature [68,69] that the purity properties of the trapping states discussed in Section 3 are preserved for perturbations of the *q* parameter. In other words, it has been shown analytically that the micromaser reaches a pure steady state whenever the θ value only is fine tuned, *irrespective* on the value of *q*:(14)$$\theta =\frac{Q}{\sqrt{m}}\;\pi ,\;\forall \;Q,m\in N\;\;\mathrm{and}\;\;\forall q\in \left[0,1\right]\;.$$

On the other hand, the robustness of such steady states for perturbations of the θ value has been discussed only recently in Ref. [25]. In view of the discussion conducted in the previous section, this is the most interesting case to consider, since the steady state itself is fragile for perturbations of the θ parameters in the incoherent case c=0. In Figure 3 and Figure 4, we show data for energy and purity during the charging protocol of the micromaser, when the incoming qubits are fully coherent, that is, c=1.

Rather remarkably, and in sharp contrast with the incoherent case c=0, we observe that the system reaches long-lived states for all the values of θ and *q*, and such states have the wanted property of being pure up to our best numerical precision [25]. In other words, we see that thanks to the coherent charging protocol, our quantum battery reaches an almost steady state which is self-sustained by its own dynamics and from which all the energy stored can in principle be extracted, since it is effectively a pure state. While such state may be a metastable one [25], our extensive simulations still show their stability up to the time scales accessible to numerical investigations.

Another important feature to discuss of this charging protocol is that the time scale to saturate the energy in the battery coincides with the time scale to reach the desired unit purity. In the incoherent (fine-tuned) case, instead, these two time scales are separated by essentially an order of magnitude (see Figure 1, where the energy saturates after about 300 collisions, and Figure 2, where the unit purity requires almost 1000 collisions to be achieved).

On the other hand, we observe that the final energy is significantly lower when compared to the incoherent setup. The reason for this reduction in the stored energy can be studied analytically for the case of θ fine tuned ($\theta =\frac{\pi }{\sqrt{m}}$), discussed in [68]. In this case, the populations $\rho_{n}$ of the steady state satisfy the following set of recursion relations:(15)$$\rho_{n}=\frac{1-q}{q}\cot^{2}\left(\frac{\pi }{2\sqrt{m}}\sqrt{n}\right)\rho_{n-1}\;,\;\forall n<m\;,$$
from which one can observe that contrary to the incoherent case, the additional *n*-dependent terms $\cot^{2}\left(\frac{\pi }{2\sqrt{m}}\sqrt{n}\right)$ suppress the high-energy populations and consequently reduces the energy stored in the battery. In the inset of Figure 3 we show how the energy stored in the steady state depends on the value of *q*, for the fine-tuned case and by using Equation (Equation 15). As expected, for q≈0 the energy approaches the value obtained in the incoherent case and it becomes reduced by increasing *q* until it tends to vanish for q≈1.

## 5. Lossy Cavity

In this section, we analyze how much the features uncovered so far for the coherent charging protocol (c=1) are stable when considering dissipative effects in the micromaser. To this purpose, we consider the following master equation, where Equation (Equation 7) is modified by adding a dissipative term [62]
(16)$$\begin{array}{c} \rho_{B}\left(k+1\right)=e^{Lt_{r}}{\mathrm{Tr}}_{q}\left[{\hat{U}}_{I}\left(\tau \right)\left(\rho_{B}\left(k\right)⊗\rho_{q}\right){\hat{U}}_{I}^{†}\left(\tau \right)\right], \end{array}$$
where $t_{r}$ denotes the time interval between consecutive collisions, assumed to be much larger than the duration τ of a single collision, so that at most a single qubit at a time interacts with the cavity. We also assume that dissipation takes place only between two collisions. Such an assumption is reasonable whenever the condition $t_{r}\gg \tau$ is satisfied. For the sake of simplicity, we have assumed the constant dissipation rate γ to be sufficiently small (γτ≪1) to allow for a single-step Trotterization of collision and dissipation, this latter ruled by the Lindbladian L. The dissipation of the cavity field is described by the master equation for a damped harmonic oscillator, with the Lindbladian [70]
(17)$$\begin{array}{c} L\left(\rho_{B}\right)=-\frac{\gamma }{2}\left(\bar{n}+1\right)\left[a^{†}a\rho_{B}-a\rho_{B}a^{†}\right]-\frac{\gamma }{2}\bar{n}\left[\rho_{B}aa^{†}-a^{†}\rho_{B}a\right]+H.c.\;. \end{array}$$

In the equation above, $\bar{n}$ denotes the mean number of thermal photons inside the cavity at a given temperature, and γ stands for the decay rate of the mean number of photons in the cavity field toward $\bar{n}$.

It is worth noting that in the fine-tuned case, $\theta =\frac{\pi }{\sqrt{m}}$, it was observed numerically in [71] that, although the separation of the phase space in disconnected sectors is no longer at play in the dissipative case, still the model exhibits long-lived states resembling their non-dissipative counterparts. In particular, it was observed that such states are almost, but not completely, pure.

We have numerically solved Equation (Equation 16) for $\gamma t_{r}\ll 1$ and $\bar{n}=0.15$. In Figure 5, we show the time evolution of the energy stored in the battery in the presence of dissipation for different values of θ. As we see, for a low decay rate, $\gamma t_{r}=10^{-3}$, the system reaches a stable value of the energy which is almost the same as the energy stored in the absence of dissipation. For a high decay rate, $\gamma t_{r}=0.1$, the system again reaches a stable value of the energy, but with lower energy and charging power compared to the energy stored in the non-dissipative case. As a matter of fact, one can also note in Figure 6 that for the case $\gamma t_{r}=10^{-3}$ the final state is almost pure and stable under its own dynamics. Furthermore, for the case $\gamma t_{r}=0.1$, purity drops to a lower value, P≈0.85, but its stability is preserved.

The results of this section show that, even in the presence of a certain degree of dissipation, the qualitative features of the charging protocol in the presence of coherences are preserved, with the final state being essentially pure and stable under time evolution. These findings reinforce the idea that micromasers can be used to practically achieve quantum batteries, displaying excellent performances in terms of the stability of the accumulated energy and of the amount of energy that, in principle, be extracted under unitary operations.

Finally, we investigate the statistics of the field in the lossy cavity. In order to characterize the nature of photon statistics, we use the Fano factor defined as
(18)$$\begin{array}{c} F\left(k\right)=\frac{\sigma^{2}\left(k\right)}{\mu (k)}, \end{array}$$
where $\sigma^{2}\left(k\right)$ and μ(k) are the variance and the mean of the photon distribution after *k* collisions. For F>1, F=1, and F<1 the cavity field has super-Poissonian, Poissonian, and sub-Poissonian statistics. Figure 7 shows the Fano factor as a function of the number of collisions, for the lossy cavity. We see that the state of the cavity field for different values of the decay rate follows the sub-Poissonian statistics, which is a property of non-classical fields.

## 6. Conclusions

In this work, we have further investigated the proposal of Ref. [25] that a micromaser, charged by means of coherent qubits, works as a very reliable quantum battery system. In contrast with incoherent charging, an effectively steady state, stable up to the times accessible to numerical investigations, is also obtained for generic values of θ=gτ. In contrast, in the incoherent case, for a generic and non fine-tuned value of $\theta \neq \pi /\sqrt{m}$, the battery absorbs an unlimited amount of energy, which would eventually “burn it up”. We have also shown that our results are stable up to moderate values of battery damping, resulting in only a moderate reduction in energy and purity for the numerically observed steady state.

Our analysis has been limited to the Jaynes–Cummings regime, which naturally features in the original micromaser experiments [62]. On the other hand, one can address the strong or even ultra-strong coupling regime, where the qubit–cavity interaction energy becomes comparable, or can even exceed the bare frequencies of the uncoupled systems [72,73], and in such regime counter-rotating terms in the interaction Hamiltonian should also be considered with care. Such terms can be handled using a simultaneous modulation of cavity and qubit frequency, as discussed in the literature [63]. However, such modulation requires a further control knob, and the stability of battery charging under unavoidable imperfections in the modulation protocol should be investigated. Therefore, it would be interesting to investigate the working of the battery including counter-rotating terms, together with cavity losses that may stabilize the battery even in this case. A careful examination of the strategies outlined above is worthwhile, since the ultra-strong coupling regime in principle offers the possibility of speeding up battery charging, and in general quantum operations.

In perspective it would be interesting to inspect solid-state platforms for micromaser-based QBs, exploiting semiconducting double quantum dot geometry [74] or superconducting quantum circuits [75], to compare the performance of our proposed model with that of early experimental proof-of-principle implementations of quantum batteries [76,77,78,79,80].

On a more fundamental ground, it would be interesting to consider a network of cavities and see whether quantum correlations between the batteries may speed up the charging process. Moreover, it is crucial to design suitable discharge protocols to rapidly and reliably extract the energy stored in the battery.

## Figures and Tables

**Figure 1 entropy-25-00430-f001:**
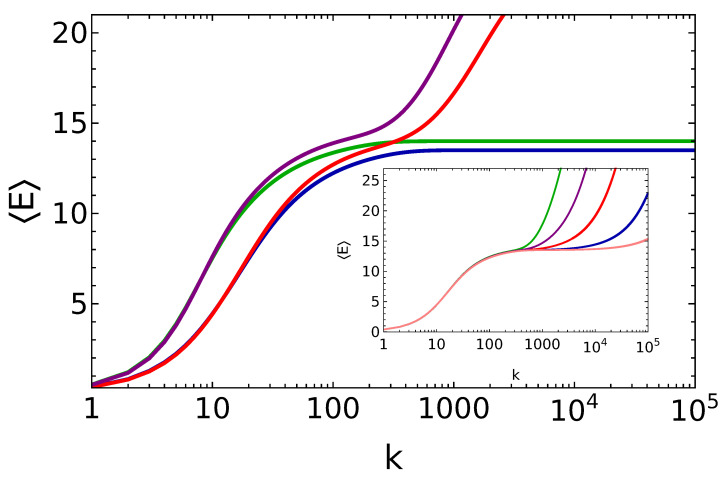
Energy of the battery vs. the number of battery–qubit collisions, for q=0.25 and $\theta =\frac{\pi }{\sqrt{15}}$ (blue line), q=0.25 and $\theta =\frac{\pi }{\sqrt{15.6}}$ (red line), q=0 and $\theta =\frac{\pi }{\sqrt{15}}$ (green line:), q=0 and $\theta =\frac{\pi }{\sqrt{15.6}}$ (purple line). Inset: Energy of the battery vs. the number of battery–qubit collisions, for q=0.25 and $\theta =\frac{\pi }{\sqrt{15.01}}$ (pink line), $\theta =\frac{\pi }{\sqrt{15.02}}$ (blue line), $\theta =\frac{\pi }{\sqrt{15.05}}$ (red line), $\theta =\frac{\pi }{\sqrt{15.1}}$ (purple line), $\theta =\frac{\pi }{\sqrt{15.2}}$ (green line).

**Figure 2 entropy-25-00430-f002:**
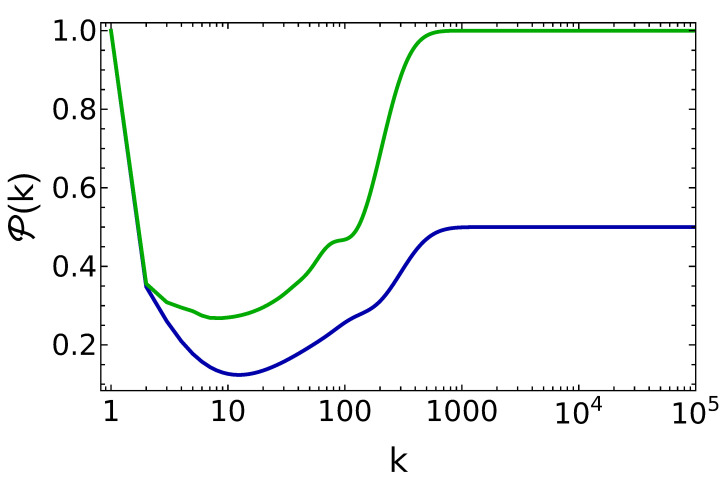
Purity of the battery vs. the number of battery–qubit collisions, for the fine-tuned value $\theta =\frac{\pi }{\sqrt{15}}$, q=0.25 (blue line) and q=0 (green line).

**Figure 3 entropy-25-00430-f003:**
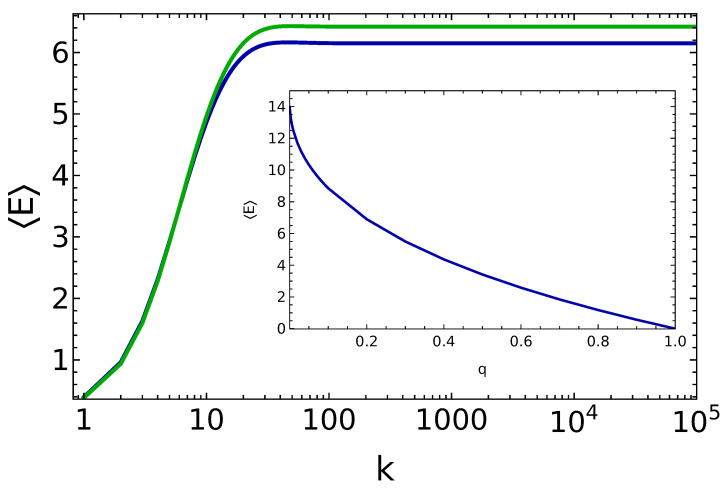
Energy of the battery vs. the number of battery–qubit collisions, for q=0.25, $\theta =\frac{\pi }{\sqrt{15}}$ (blue line) and $\theta =\frac{\pi }{\sqrt{15.6}}$ (green line). Inset: the steady-state energy for the case $\theta =\frac{\pi }{\sqrt{15}}$ as a function of the parameter *q*, as computed by means of Equation (Equation 15).

**Figure 4 entropy-25-00430-f004:**
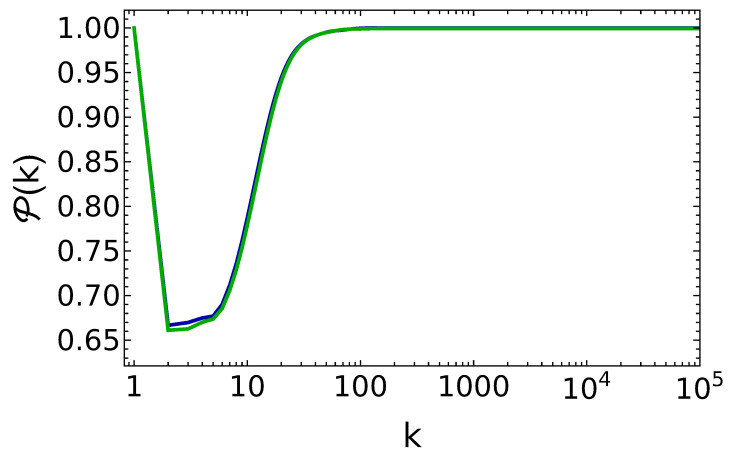
Purity of the battery vs. the number of battery–qubit collisions, for the same parameter values as in Figure 3.

**Figure 5 entropy-25-00430-f005:**
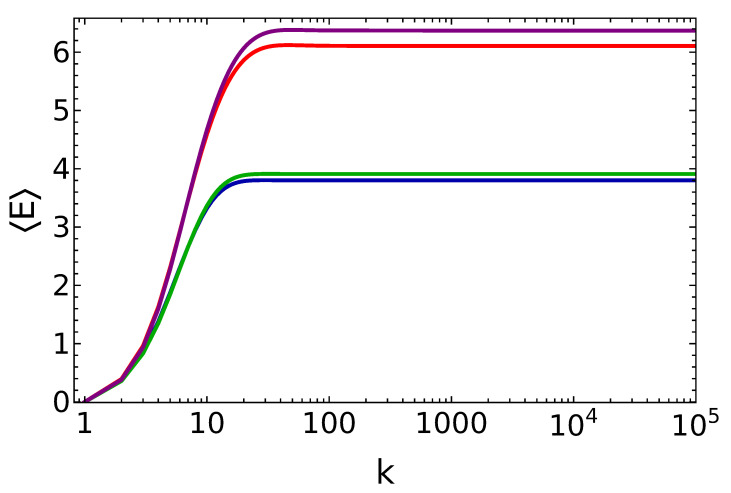
Energy of the battery vs. the number of battery–qubit collisions, in the presence of dissipation, at q=0.25 and $\bar{n}=0.15$, $\theta =\frac{\pi }{\sqrt{15}}$ and $\gamma t_{r}=0.1$ (blue line), $\theta =\frac{\pi }{\sqrt{15.6}}$ and $\gamma t_{r}=0.1$ (green line), $\theta =\frac{\pi }{\sqrt{15}}$ and $\gamma t_{r}=0.001$ (red line), $\theta =\frac{\pi }{\sqrt{15.6}}$ and $\gamma t_{r}=0.001$ (purple line).

**Figure 6 entropy-25-00430-f006:**
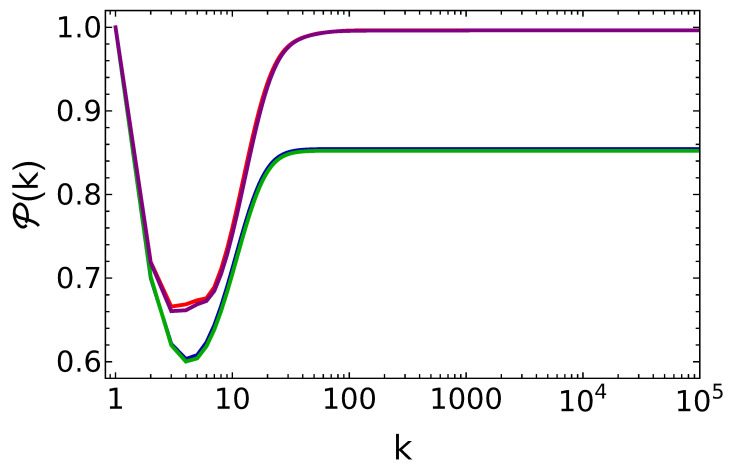
Purity of the battery vs. the number of battery–qubit collisions, for the same parameter values as in Figure 5. Notice the almost perfect overlap of curves in pairs.

**Figure 7 entropy-25-00430-f007:**
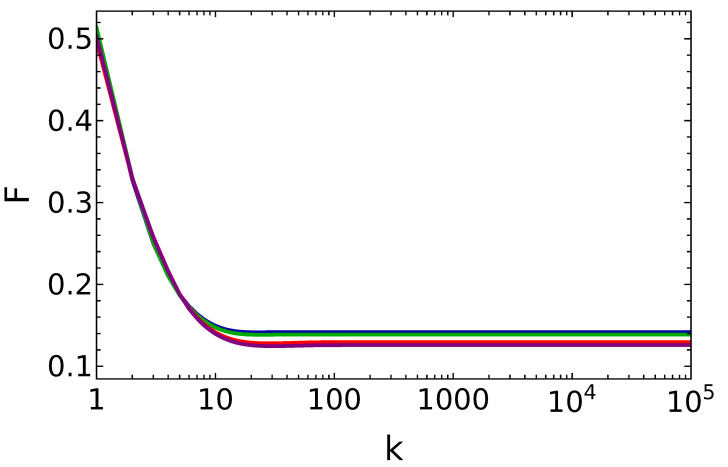
Fano factor vs. the number of battery–qubit collisions, in the presence of dissipation, for q=0.25 and $\bar{n}=0.15$, $\theta =\frac{\pi }{\sqrt{15}}$ and $\gamma t_{r}=0.1$ (blue line), $\theta =\frac{\pi }{\sqrt{15.6}}$ and $\gamma t_{r}=0.1$ (green line), $\theta =\frac{\pi }{\sqrt{15}}$ and $\gamma t_{r}=0.001$ (red line), $\theta =\frac{\pi }{\sqrt{15.6}}$ and $\gamma t_{r}=0.001$ (purple line). Notice the almost perfect overlap between pairs of curves.

## Data Availability

The data are contained within the article.

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
