# Peer review of "Lossy Micromaser Battery: Almost Pure States in the Jaynes–Cummings Regime"

_entropy, 2023, doi:10.3390/e25030430_

Round 1

Reviewer 1 Report

This article investigates the energy charging process of a cavity by atoms in the micromaser setup. It expands previous work by the group by considering the effects of losses in the cavity due to the coupling with environmental radiation. It also discusses the situation in which the incoming atoms are incoherent.  

In the incoherent case, the steady state is given analytically, and numerical results are presented to discuss the charging process as a function of time dependence (or collision number). One observes that for values of theta different from $\pi/\sqrt{m}$, with an integer $m$, the energy blows up. However, there is a transient saturation at approximately 100 collisions for the cyan and red line in Figure 1. Is there some intuitive understanding of what happens there? Is this "transient saturation region" larger for $x=15.01$, for instance, that for $x=15.6$ as chosen for the plot (with $\theta=\pi/\sqrt{x}$)? It will be nice to have some comments about that. 

In general terms, the results of the article are interesting. The presentation is clear and concise. The introduction provides a good review of the previous literature—however, the paper "Dissipative charging of a quantum battery" Phys. Rev. Lett. 122, 210601 (2019) appeared earlier than the references [13-16,24] on collisional models to describe the charging process of a quantum battery and deserves to be mentioned there.  

Author Response

We thank the reviewer for the positive evaluation of our work and constructive comments. 

Comment:
"In the incoherent case, the steady state is given analytically, and numerical results are presented to discuss the charging process as a function of time dependence (or collision number). One observes that for values of theta different from $\pi/\sqrt{m}$, with an integer $m$, the energy blows up. However, there is a transient saturation at approximately 100 collisions for the cyan and red line in Figure 1. Is there some intuitive understanding of what happens there? Is this "transient saturation region" larger for $x=15.01$, for instance, that for $x=15.6$ as chosen for the plot (with $\theta=\pi/\sqrt{x}$)? It will be nice to have some comments about that."

Our answer:
The transient saturation regime becomes longer when approaching a fine tuned value. Specific values 
for the case of Fig. 2 have been added to the text. Of course there would be a cost of precision in 
any experimental implementation that tries to approximate so closely fine-tuned values. 

Comment;
"In general terms, the results of the article are interesting. The presentation is clear and concise. The introduction provides a good review of the previous literature—however, the paper "Dissipative charging of a quantum battery" Phys. Rev. Lett. 122, 210601 (2019) appeared earlier than the references [13-16,24] on collisional models to describe the charging process of a quantum battery and deserves to be mentioned there."

Our answer:
We thank the reviewer for bringing this paper to our attention. We have added it to our reference list (new Ref.[24])  

Reviewer 2 Report

The authors study a model of a quantum battery in which a harmonic mode is charged by repeated interactions with a collection of two-level systems. They investigate the existence and properties of the steady state of the harmonic oscillator. Particular emphasis is placed on the purity of this steady state, as it may enable more efficient energy extraction.

The paper is well written, and of some interest. However, it suffers from a few problems that must be addressed.

·         The main phenomenology, namely the creation of a pure steady state by incoherent driving, or in the fully coherent case, seem to have been discussed in previous publications, including by the authors. What is added here seems to be a straightforward numerical investigation of a few representative cases, as well as adding some measure of thermalization to the (formerly) coherent model. I am not convinced that this is sufficient.

·         In my opinion a more in depth discussion of the details, which goes beyond the immediate presentation of the numerical results, is needed. I will give two examples.

-       In Sec. 3 it is clear that the parameter q is chosen such that the steady state energy is 14 (or a close value, depending on q). For somewhat offset variable q the system approximates a similar state, but later escapes towards higher energies. It should be possible to analytically estimate the typical time to reach the steady state, as well as the subsequent escape time, as a function of parameters.

-          In Sec. 4 the authors emphasize the fact that the steady state is stable for coherent driving. They fail to mention that the mean energy in the example shown is very far from the value of Sec. 3 (E=14), meaning that the previous mechanism is not really relevant here. They should try to estimate how Eq. (15) determines the mean energy of the steady state. Then one can study how taking q to 0 affects this steady state. I expect that only for sufficiently low values of q the value of the parameter q becomes important.

One final minor point, in some of the figures the colors of the lines are rather similar (e.g. cyan and blue). I would recommend selecting more distinct colors, and perhaps making some of the lines dashed or dotted.

To summarize, the paper includes some interesting results, but I am not convinced they are sufficient to justify a new publication. I therefore cannot recommend publication of the manuscript in its current form.

Author Response

Comment:
"The authors study a model of a quantum battery in which a harmonic mode is charged by repeated interactions with a collection of two-level systems. They investigate the existence and properties of the steady state of the harmonic oscillator. Particular emphasis is placed on the purity of this steady state, as it may enable more efficient energy extraction.

The paper is well written, and of some interest."

Our answer:
We thank the reviewer for their encouraging comments.

Comment:
"However, it suffers from a few problems that must be addressed.

·         The main phenomenology, namely the creation of a pure steady state by incoherent driving, or in the fully coherent case, seem to have been discussed in previous publications, including by the authors. What is added here seems to be a straightforward numerical investigation of a few representative cases, as well as adding some measure of thermalization to the (formerly) coherent model. I am not convinced that this is sufficient."

Our answer:
It is true that we compare here coherent and incoherent charging of the battery. Nevertheless, this is not the main novelty element of our paper. The main new point here is to include in the analysis cavity losses, while previous analysis was purely Hamiltonian. We show that battery performances, in terms of stored energy, charging power, and steady-state purity, are slightly degraded up to moderated dissipation rate. This stability analysis is a crucial and necessary step in view of possible future experimental implementations. 

Comment:
"In my opinion a more in depth discussion of the details, which goes beyond the immediate presentation of the numerical results, is needed. I will give two examples.

-       In Sec. 3 it is clear that the parameter q is chosen such that the steady state energy is 14 (or a close value, depending on q). For somewhat offset variable q the system approximates a similar state, but later escapes towards higher energies. It should be possible to analytically estimate the typical time to reach the steady state, as well as the subsequent escape time, as a function of parameters."

Our answer:
The escape discussed in Fig. 3 is not as a function of q but of theta, so we assume the reviewer refers to this parameter. Only for fine-tuned values of theta a steady-state exists. To address the dependence of the escape time on the deviation (Delta theta) of the parameter theta from a fine-tuned value, we have added an inset in Fig. 1. Our understanding, explained in the text (new sentences highlighted in red) is that such deviation introduces an effective random noise to the steady-state fine-tuned energy determined by Eqs. (9) and (10), which becomes relevant at time t^star \propto 1/(Delta\theta)^2, an estimate consistent with the inset of Fig. 1.

Comment:
"-          In Sec. 4 the authors emphasize the fact that the steady state is stable for coherent driving. They fail to mention that the mean energy in the example shown is very far from the value of Sec. 3 (E=14), meaning that the previous mechanism is not really relevant here. They should try to estimate how Eq. (15) determines the mean energy of the steady state. Then one can study how taking q to 0 affects this steady state. I expect that only for sufficiently low values of q the value of the parameter q becomes important."

Our answer: 
Both in the coherent and in the incoherent case the steady-state is obtained since the full Hilbert space splits in chambers, dynamically separated from each other. So, the mechanism leading to the presence of a steady state is the Hilbert Space splitting and it is exactly the same mechanism in coherent and incoherent driving. The difference between the two cases is that, in the case of coherent driving, the non perfect fine tuning of the theta parameter does not completely spoil the Hilbert space splitting property, thus leading to an effectively steady state which is not present in the incoherent case. In the coherent case the steady-state has lower energy than in the incoherent case since several number states are populated, see Eq. (15) and not just a single number state, as in Eq. (10). To address the reviewer's question, we have added an inset to Fig.3, where, using Eq. (15), we plot the steady-state energy as a function of q. We can see that the value <E>=14 of the energy is recovered for q=0, where the chargers are prepared in the excited state |e>, for which the density matrix is diagonal, and so the incoherent result (10) applies. The steady-state energy then reduces with q, as expected, but quite high values are obtained up to values of q not small, for instance the energy is equal to about a half of the q=0 energy at q=0.2.

Comment:
"One final minor point, in some of the figures the colors of the lines are rather similar (e.g. cyan and blue). I would recommend selecting more distinct colors, and perhaps making some of the lines dashed or dotted."

Our answer:
We have chosen a better color code and improved the quality of our figures.

Comment:
"To summarize, the paper includes some interesting results, but I am not convinced they are sufficient to justify a new publication. I therefore cannot recommend publication of the manuscript in its current form."

Our answer:
We have clarified all points raised by the referee and therefore hope that our paper can now be accepted for publication.

Reviewer 3 Report

In this work the authors investigate the transfer of energy of a stream of qubits into a cavity. The term such a device a micromaser battery. They compare the coherent and incoherent cases, find the former to be more efficient in energetic transference (as one would expect). This work scientifically sound and sufficiently novel for publication in Entropy. I have just one suggestion: in the Introduction section where the authors discuss prior work, they fail to discuss the recent spate of experimental work in quantum batteries. I would suggest including a discussion on this experimental work, of which there are now a several: e.g. Quach, et al. Sci. Adv. 2022, 8, eabk3160; Gemme, et al. Batteries 2022, 8, 43; Hu, et al. arXiv:2108.04298; De Buy Wenniger, et al. arXiv:2202.01109; Joshi et al. arXiv:2112.1543.

Author Response

We thank the Referee for the positive evaluation of our paper and for the suggested references. Indeed we have added a sentence in our conclusions referring to early experimental implementations of quantum batteries, quoting new Refs. [77]-[81]. 

Round 2

Reviewer 1 Report

The authors have made the appropriate changes. 

Reviewer 3 Report

The manuscript is suitable for publication in present form.